# Multi-year comparison of VITEK MS performance for identification of rarely encountered pathogenic gram-positive organisms (GPOs) in a large integrated Canadian healthcare region

D.L. Church,[1,2,3] T. Griener,[1,3] D. Gregson[1,2,3]

**ABSTRACT**  This multi-year study (2014–2019) compared identification of rare and unusual gram-positive organisms (GPOs) by matrix-assisted laser desorption ionization-time of flight mass spectrometry (MALDI-TOF MS) (VITEK MS; bioMérieux, Laval, Quebec) to 16S rRNA gene sequencing (16S) according to our laboratory routine workflow. 16S is done if initial MALDI-TOF MS results are discordant or wrong, or there are no results. GPO isolates were first analyzed by standard phenotypic methods and MALDI-TOF MS using direct deposit with full formic acid extraction; MALDI-TOF was repeated if no result occurred. Medically approved 16S analyses were done using fast protocols. Isolate sequences were analyzed using the Integrated Database Network System bacterial database (SmartGene, Lausanne, Switzerland). 655 GPO isolates were recovered from 648 specimens; >1 isolate was recovered from 7 (1%). A total of 451 (68.9%) aerobic gram-positive bacilli (GPBs) and 204 (31.1%) aerobic gram-positive cocci (GPCs) were mainly recovered from bloodstream infections (35%), sterile fluids and deep tissues (35%), and abscesses/deep wounds (17%). Accurate genus vs species identities were obtained for 59% and 49.4% GPB, and 81% and 53.9% GPC, respectively. Wrong or no results were obtained for 9% and 31% of GPB and 7% and 12% of GPC; 15% of GPBs and 5.3% of GPC identification errors occurred due to absence from the instrument's database. VITEK MS performance remained stable for GPB and GPC isolates due to few species additions to the database. VITEK MS databases need to be continually updated to include an increasing number of rare and unusual GPOs causing invasive human infections. 16S remains important for identification of GPOs where MALDI-TOF fails.

**IMPORTANCE** MALDI-TOF MS has transformed the identification of commonly encountered GPOs in the clinical laboratory, but rare and unusual bacteria continue to challenge the technology. This study verified the performance of VITEK MS for identification of a broad range of rare and unusual clinical GPO isolates by our large reference laboratory workflow over a multi-year period. Although most GPOs were accurately identified by MALDI-TOF MS, a small number of common GPC isolates (6.3%) (i.e., *Enterococcus*/*Staphylococcus*/*Streptococcus*) requiring sequencing for identification were studied. Approximately 13% of aerobic GPBs and 5.3% of GPCs could not be accurately identified by MALDI-TOF due to lack of an organism in the instrument's database. MALDI-TOF MS databases should be continuously updated and validated, and laboratories should have a workflow for the identification of unusual or rarely encountered GPOs that includes 16S rRNA gene sequencing whenever MALDI-TOF cannot give a definitive identification.

**KEYWORDS**  identification, gram-positive bacteria, VITEK MS, MALDI-TOF

Address correspondence to D.L. Church , dchurch@ucalgary.ca.

The authors declare no conflict of interest.

Aerobic gram-positive bacteria (GPO) are commonly isolated from various specimen types in the clinical microbiology laboratory. Both gram-positive bacilli (GPBs) and gram-positive cocci (GPCs) are part of the skin microbiome of humans and animals, and new genera and species are being annually reported (1). They are increasingly being recognized as important causes of invasive human infection. Immunocompromised patients, those with hematological malignancies, and those undergoing invasive procedures, such as intravascular catheterization, implantation of prosthetic devices, and prolonged courses of broad-spectrum antibiotic therapy, are at higher risk (2–10). Bloodstream infections caused by a wide range of aerobic GPOs require definitive identification of these opportunistic pathogens to determine the source and proceed with a sound management plan (4, 6, 11–13). Accurate identification of many aerobic GPOs remains challenging, despite the widespread implementation of newer matrix-assisted laser desorption ionization-time of flight mass spectrometry (MALDI-TOF) and genomics methods.

Definitive identification of GPOs is suboptimal using phenotypic methods alone because of low biochemical reactivity, but many reactive species also have identical or nearly identical phenotypes. Accuracy and timeliness of GPO identification, along with patient outcomes, have improved with the use of MALDI-TOF MS (2, 11, 14–16). MALDI-TOF MS databases currently lack spectral profiles for some unusual and rarely encountered pathogenic GPOs, resulting in isolate misidentification or lack of identification. Limited studies compared VITEK MS (MALDI-TOF MS) (bioMérieux, Laval, Quebec) to 16S rRNA gene sequencing (16S) for identification of unusual or rarely encountered pathogenic GPB or GPC organisms (2, 14–16). Prior reports of MALDI-TOF MS performance used GPB organisms available in MALDI-TOF databases that had prior 16S analysis to evaluate identification of *Corynebacterium* spp. (17, 18), *Actinomyces* spp. (12, 19), and *Listeria* spp. (20, 21). We verified the performance of our laboratories' workflow by comparing VITEK MS results since implementation in 2014 to secondary 16S analysis of isolates where MALDI-TOF misidentified or failed to provide a result. This study is important because few studies have verified VITEK MS (MALDI-TOF MS) (bioMérieux) performance within a sequential laboratory workflow. MALDI-TOF MS is first done with secondary analysis using 16S rRNA gene sequencing when MALDI-TOF provides a wrong identification or no result.

## MATERIALS AND METHODS

### Patients and clinical specimens

Patients who had ≥1 GPO isolate from blood or non-blood invasive clinical specimens were enrolled over a multi-year period (2014–2019) from Calgary and South Zones, Alberta Health Services, following a clinical review by a medical microbiologist and infectious disease specialist. Adults had two sets of blood cultures (i.e., each consisting of an aerobic/anaerobic bottle) according to Calgary Zone regional protocol. Non-blood specimens (abscesses, tissues, and sterile fluids) were collected operatively or by interventional radiology under ultrasound guidance. Non-blood specimens were collected using standard collection devices and protocols to ensure recovery of GPO. Clinical specimens were transported for immediate processing to the laboratory within 2 h after collection.

### Laboratory setting

Our large regional laboratory in Calgary Zone, Alberta Health Service, does clinical microbiology testing for Calgary and surrounding hamlets and towns in Southern Alberta representing an urban and rural population of ~2.6 million people, including adult tertiary hospitals, a tertiary pediatric hospital, several rural facilities, all ambulatory practices, and long-term care centers. A higher number of unusual or rarely invasive GPOs are encountered in our large referral facility compared to smaller hospital laboratories.

## Laboratory analyses

Clinical specimens were initially analyzed using standard phenotypic methods including a microscopic examination and aerobic and anaerobic culture methods (except for sputa). Blood and sterile fluids cultured using an aerobic BACT/ALERT FA plus and FN plus bottle pair were incubated for up to 5 days in a BacT/Alert instrument (bioMérieux). Subcultures of positive blood cultures were inoculated onto blood agar (BA), chocolate agar (CHOC), MacConkey agar (MAC), and *Brucella* blood agar (BBA) (Dalynn, Calgary) plates and incubated for 5 days at 35°C. Subcultures of sputum and deep lung samples (i.e., bronchoalveolar lavages and bronchial washes) were inoculated onto BA, CHOC, MAC, and buffered charcoal yeast extract agar and incubated under aerobic conditions at 35°C for 4 days. Tissue and deep abscess specimens were inoculated onto BA, CHOC, and BBA; BA and CHOC plates were incubated in 5% $CO_2$; MAC was incubated in $O_2$; and BBA was incubated in an Anoxomat anaerobic jar system (Fisher Scientific, Mississauga, Ontario) for 4 days. GPO isolates were confirmed as either aerobic gram-positive bacilli or GPCs by standard phenotypic procedures (i.e., growth on specific media, atmospheric growth conditions, colony morphology, Gram stain reaction/morphology, and rapid biochemical tests [e.g., including but not limited to catalase, coagulase, PYR, indole, CAMP test, esculin, Taxo P disk test, optochin, Bacitracin disk test, and $H^2S$]). Gram-positive bacilli and cocci included organisms that grew on BA and CHOC but not on MAC. GPB and GPC isolates were subsequently analyzed according to our laboratory's workflow; MALDI-TOF MS is first done with subsequent 16S rRNA gene sequencing when MALDI-TOF gives a low confidence or absent result. Direct deposit full formic acid extraction, proteomic analysis, and result interpretation were done for MALDI-TOF MS (VITEK MS, bioMérieux) according to manufacturer's instructions as described in previously published guidelines (22). Repeat MALDI-TOF MS was done with the same method if initial results gave low confidence (<99%), ambiguous (e.g., slashline results), or no identification as defined by the manufacturer. Reported VITEK MS results occurred where the software gave one choice with a high confidence (i.e., ≥99.0%) and agreed with phenotypic results. Two or more results with the same genus but multiple species were considered acceptable for genus identification only unless 16S sequencing provided a high-confidence identification within the first ~500 bp of 16S to the genus or species level according to previously published guidelines (23). A wrong identification occurred where the genus and species results given by MALDI-TOF MS were completely discordant with the molecular result. Food and Drug Administration-cleared VITEK MS databases were used throughout the study, not the research library. VITEK MS v.2.0 (2012) database was used from 2014 to 2016, v.3.0 from 2016 to 2018, and v.3.2 until 2019. Although the identification from the instrument databases may have provided an older taxonomic name, the most recent taxonomic change for GPOs is reflected herein with the older name designated (formerly named) according to the List of Prokaryotes with Standing in Nomenclature (lpsn.dsmz.de) (see Tables 1 and 2) (1). A total of 1,745 GPB and 26,489 GPC isolates were not enrolled because they could be routinely identified with high confidence using phenotypic plus VITEK MS analysis, and thus, 16S was not performed. These were designated as "common GPOs" for the study. All enrolled GPOs and GPCs underwent molecular analysis because there was a discrepancy between the phenotypic and MALDI-TOF results, or MALDI-TOF gave no result or a wrong identification. These were designated as "rare or unusual GPOs" for the study.

16S rRNA gene sequencing was performed by fast PCR/cycle sequencing using fast MicroSEQ 500 16S DNA PCR kits and an ABI Prism 3500 XL sequencer (Applied Biosystems and Thermo Fisher Scientific, Foster City, CA), as previously described (25). SmartGene's Integrated Database Network System (Lausanne, Switzerland) bacterial database was used to determine closely related species (https://www.Smartgene.com). SmartGene continuously updates its bacterial and centroid databases. Isolate sequences included in the study were compared to a well-characterized reference sequence, and overall identity scores were 99.9% (zero to two mismatches). 16S rRNA gene sequencing

**TABLE 1** Performance of VITEK MS compared to molecular identification of aerobic gram-positive bacilli[c]

| Reference method | VITEK MS results | | | | | Total |
|---|---|---|---|---|---|---|
| 16S RNA gene sequencing result[d,e] | No. (%) of correct identifications at genus level | No. (%) of correct identifications at genus and species levels | No. (%) with discordant species results[b] | No. (%) with wrong result[b] | No. (%) with no results | Total no. (%) |
| *Actinobacterium* sp. | | _[g] | | | 2 | 2 |
| *Arcanobacterium haemolyticum* | 11 | 11 | | | | 11 |
| *Actinomyces gerencseriae* | 1 | 1 | | 1 (*Corynebacterium durans*) | 2 | 4 |
| *Actinomyces graevenitzii* | 2 | 2 | | 1 (*Propionibacterium prionioncum*) | 1 | 4 |
| *Actinomyces israelii* | 3 | 3 | | | 6 | 9 |
| *Actinomyces urogenitalis* | | | | | 1 | 1 |
| *Actinomyces viscosus* | 3 | 3 | | | | 3 |
| *Actinomyces* sp. | 11 | – | 6 (*Actinomyces viscosus* [6]) | 5 (*Staphylococcus warneri* [1], *Sphingomonas paucimobilis* [1], *Campylobacter coli* [1], *Kocuria kristinae* [1], *Parvomonas micra* [1]) | 8 | 24 |
| *Atopobium* sp. | | – | | 1 (*Parvomonas micra*) | 1 | 2 |
| *Arthrobacter* sp. | 2 | – | 1 (*Arthrobacter cumminsii*) | 1 (*Brucella* sp.) | 4 | 7 |
| ***Bacillus aerophilus*** | 1 | | 1 (*Bacillus pumilis*) | | | 1 |
| *Bacillus cereus* | 1 | 1 | | | 2 | 3 |
| *Bacillus cereus/Bacillus thuringiensis* | 1 | | 1 (*Bacillus cereus/B. thuringiensis*) | | | 1 |
| *Bacillus thuringiensis* | 1 | | 1 (*B. cereus/Bacillus mycoides/ thuringiensis*) | | | 1 |
| *Bacillus* sp. | 3 | – | 1 (*Bacillus cereus*) | | 9 | 12 |
| ***Bifidobacterium adolescentis*** | 1 | | | | | 1 |
| *Bifidobacterium breve* | 9 | 6 | | | 2 | 11 |
| *Bifidobacterium dentium* | 1 | | | | | 1 |
| *Bifidobacterium longum* | 5 | 3 | | | 1 | 6 |
| *Bifidobacterium* sp. | 4 | – | | | 1 | 5 |
| *Brevibacillus agri* | 1 | | | | 1 | 2 |
| *Brevibacillus brevis* | 2 | | | | | 2 |
| ***Brevibacterium paucivorans*** | | | | 1 (*Bacteroides ovatus/Bacteroides xylanisolvens*) | | 1 |
| *Brevibacterium casei* | 4 | 4 | | | | 4 |
| *Brevibacterium luteolum* | 2 | 2 | | | | 2 |
| *Brevibacterium* sp. | 3 | – | 3 (*Brevibacterium casei*) | | | 3 |
| *Carnobacterium* sp. | | – | | | 1 | 1 |
| *Cellulosimicrobium cellulans* | | | | | 2 | 2 |
| *Collinsella aerofaciens* | | | | | 1 | 1 |
| ***Coprobacillus cateniformis*** | | | | | 1 | 1 |
| *Corynebacterium accolens* | 2 | | 2 (*C. diphtheriae* and *C. tuberculostearicum*) | | | 2 |
| *Corynebacterium amycolatum* | 4 | | 4 (*C. amycolatum/ Corynebacterium xerosis*) | | | 4 |
| *Corynebacterium aurimucosum* | 4 | 3 | 1 (*C. tuberculostearicum*) | | | 4 |
| *Corynebacterium bovis* | 1 | 1 | | | | 1 |
| ***Corynebacterium canis*** | | | | | 2 | 2 |
| *Corynebacterium diphtheriae*[f] | 2 | 2 | | | | 2 |
| *Corynebacterium durum* | | | | | 2 | 2 |
| ***Corynebacterium freiburgense*** | | | | | 1 | 1 |

(*Continued on next page*)

**TABLE 1** Performance of VITEK MS compared to molecular identification of aerobic gram-positive bacilli$^c$ (Continued)

| Reference method | VITEK MS results | | | | | Total |
|---|---|---|---|---|---|---|
| 16S RNA gene sequencing result$^{d,e}$ | No. (%) of correct identifications at genus level | No. (%) of correct identifications at genus and species levels | No. (%) with discordant species results$^b$ | No. (%) with wrong result$^b$ | No. (%) with no results | Total no. (%) |
| *Corynebacterium jeikeium* | 4 | 2 | 2 (*C. amylolatum* and *C. amylolatum/C. xerosis*) | | | 4 |
| *Corynebacterium kroppenstedtii* | 11 | 10 | 1 (*Corynebacterium minutissimum*) | | 4 | 15 |
| *Corynebacterium macginleyi* | 2 | 2 | | | 1 | 3 |
| *Corynebacterium massiliense* | | | | | 1 | 1 |
| *Corynebacterium matruchotii* | 1 | 1 | | | | 1 |
| *Corynebacterium mucifaciens* | 2 | 2 | | | | 2 |
| *Corynebacterium riegelii* | | | | | 1 | 1 |
| *Corynebacterium striatum* | 2 | 2 | | | | 2 |
| *Corynebacterium tuberculostearicum* | 10 | 10 | | | 2 | 12 |
| *Corynebacterium ulcerans* | 1 | | 1 (*Corynebacterium pseudodiphtheriticum*) | | | 1 |
| *Corynebacterium urealyticum* | | | | 1 (*Staphylococcus epidermidis*) | | 1 |
| *Corynebacterium* sp. | 12 | – | 11 (*C. accolens* [1], *C. striatum* [1], *C. tuberculostearicum* [6], *C. amylolatum/C. xerosis* [2], and *Corynebacterium propinquum* [1]) | 5 (*Actinomyces naeslundii* [2], *Bifidobacterium* sp. (1), *Streptococcus salivarius* [1], *Propionibacterium avidum* [1], and *Kocuria kristinae* [1]) | 9 | 26 |
| *Dermabacter hominis* | 28 | 28 | | | | 28 |
| *Dermabacter jinjuensis* | 5 | | 3 (*D. hominis*) | 1 (*Helcococcus kunzii*) | 1 | 7 |
| *Dermabacter* sp. | 2 | – | 2 (*D. hominis*) | | | 2 |
| *Dermatophilus congolensis* | 1 | 1 | | | | 1 |
| *Erysipelothrix rhusiopathiae* | 1 | 1 | | | | 1 |
| *Fannyhessea vaginae* (formerly *Atopobium vaginae*) | 1 | 1 | | 1 (*Listeria graya*) | 2 | 4 |
| *Gardnerella vaginalis* | 7 | 7 | | | 1 | 8 |
| *Gleimia europaea* (formerly *Actinomyces europaeus*) | 1 | | 1 (*Schaalia radingae*) | 1 (*Mycobacterioum kansasii*) | 4 | 6 |
| **Helcobacillus massiliensis** | | | | | 1 | 1 |
| *Janibacter* sp. | | | | | 2 | 2 |
| **Kribbia dieselivorans** | | | | | 1 | 1 |
| **Kroppenstedtia eburnea** | | | | | 1 | 1 |
| **Lacticaseibacillus paracasei** (formerly *Lactobacillus paracasei*) | 1 | | 1 (*Lacticaseibacillus casei*/L. paracasei/*L. rhamnosus* split) | | | 1 |
| **Lacticaseibacillus rhamnosus** (formerly *Lactobacillus rhamnosus*) | 1 | | 1 (*L. casei/L. paracasei/L. rhamnosus* split) | | 1 | 2 |
| **Lacticaseibacillus zeae** (formerly *Lactobacillus zeae*) | 1 | | 1 (*L. casei/L. paracasei* split) | | | 1 |
| *Lactiplantibacillus plantarum* (formerly *Lactobacillus plantarum*) | 3 | | 3 (*L. plantarum/Lactiplantibacillus paraplantarum* split) | | | 3 |
| **Lactobacillus acidophilus** | | | | | 1 | 1 |
| *Lactobacillus crispatus* | | | | 1 (*Gardnerella vaginalis*) | | 1 |
| **Lactobacillus gasseri** | 1 | | 1 (*L. gasseri/L. acidophilus* split) | 1 (*Nocardia* sp.) | 1 | 3 |

*(Continued on next page)*

**TABLE 1** Performance of VITEK MS compared to molecular identification of aerobic gram-positive bacilli[c] (Continued)

| Reference method | VITEK MS results | | | | | Total |
|---|---|---|---|---|---|---|
| 16S RNA gene sequencing result[d,e] | No. (%) of correct identifications at genus level | No. (%) of correct identifications at genus and species levels | No. (%) with discordant species results[b] | No. (%) with wrong result[b] | No. (%) with no results | Total no. (%) |
| Lactobacillus iners | | | | 1 (Gardnerella vaginalis) | | 1 |
| Lactobacillus sp. | 2 | – | 1 (Latilactobacillus sakei) | | | 1 |
| Lancefieldella parvula (formerly Atopobium parvula) | 4 | 4 | | 2 (Cardiobacterium divergens and Eggerthella lenta) | 1 | 7 |
| Lancefieldella rimae (formerly Atopobium rimae) | 2 | 2 | | | 2 | 4 |
| **Ligilactobacillus animalis (formerly Lactobacillus animalis)** | | | | 2 (Streptococcus vestibularis [1] and Cronobacter dublensis [1]) | 1 | 3 |
| Ligilactobacillus salivarius (formerly Lactobacillus salivarius) | 1 | 1 | | | | 1 |
| **Limosilactobacillus vaginae (formerly Lactobacillus vaginalis)** | | | | 1 (Corynebacterium durans) | | 1 |
| Listeria monocytogenes | 7 | 7 | | | | 7 |
| **Lysinibacillus halotolerans** | 1 | | 1 (Lysinibacillus fusiform) | | | 1 |
| Microbacterium aurum | | | | 1 (Paenibacillus durans) | | 1 |
| **Microbacterium foliorum** | 1 | | 1 (M. oxydans) | | | 1 |
| **Microbacterium lacticum** | | | | 1 (Paenibacillus durans) | 1 | 2 |
| Microbacterium oxydans | 2 | 2 | | | | 2 |
| Microbacterium sp. | 1 | – | 1 (Microbacterium paradoxyans) | 1 (Paenibacillus durans) | 3 | 5 |
| **Niallia circulans (formerly Bacillus circulans)** | 1 | 1 | | | | 1 |
| **Nialla nealsonni (formerly Bacillus nealsonni)** | | | | | 1 | 1 |
| **Oceanobacillus massiliensis** | | | | | 1 | 1 |
| **Paeniarthrobacter nicotinovorans (formerly Arthrobacter nicotinovorans)** | | | | 1 (Microbacterium arborescens) | 1 | 2 |
| **Paenibacillus anaericanus** | | | | | 1 | 1 |
| **Paenibacillus motobuensis** | | | | | 1 | 1 |
| Paenibacillus provencensis | 1 | 1 | | | 1 | 2 |
| **Paenibacillus wynnii** | | | | | 1 | 1 |
| Paenibacillus sp. | 6 | – | 3 (Paenibacillus pabuli [2] and P. provencensis [1]) | 2 (Prevotella oris) | 1 | 9 |
| **Paenisporosarcina quisquilarium** | | | | | 1 | 1 |
| Peribacillus simplex (formerly Bacillus simplex) | | | | | 1 | 1 |
| **Promicromonospora kroppenstedtii** | | | | | 1 | 1 |
| **Pseudoglutaminibacter cumminsii (formerly Arthrobacter cumminsii)** | 1 | 1 | | | 1 | 2 |
| **Rebacterium sp.** | | | | 1 (Nocardia nova/Nocardia africana) | | 1 |
| **Rhodococcus equi** | | | | 1 (Rhodococcus equi/Nocardia asteroides) | | 1 |
| Rhodococcus sp. | | | | 1 (Paenibacillus durans) | | 1 |
| Rothia aeria | 1 | 1 | | | 8 | 9 |
| Rothia dentocariosa | 6 | 6 | | 2 (Rothia mucilaginosa) | 1 | 9 |

**TABLE 1** Performance of VITEK MS compared to molecular identification of aerobic gram-positive bacilli[c] (*Continued*)

| Reference method | VITEK MS results | | | | | Total |
|---|---|---|---|---|---|---|
| 16S RNA gene sequencing result[d,e] | No. (%) of correct identifications at genus level | No. (%) of correct identifications at genus and species levels | No. (%) with discordant species results[b] | No. (%) with wrong result[b] | No. (%) with no results | Total no. (%) |
| *Rothia koreenensis* (formerly *Kocuria koreensis*) | | | | 1 (*Paenibacillus* sp.) | 1 | 2 |
| *Rothia kristinae* | 2 | 2 | | | | 2 |
| *Rothia mucilaginosa* | 5 | 5 | | | | 5 |
| *Rothia* sp. | 2 | – | 2 (*Rothia mucilaginosa*) | | | 2 |
| *Schaalia georgiae* (formerly *Actinomyces georgiae*) | | | | | 1 | 1 |
| *Schaalia meyeri* (formerly *Actinomyces meyeri* | | | | | 3 | 3 |
| *Schaalia odontolytica* (formerly *Actinomyces odontolyticus*) | 6 | 3 | 3 (*Actinomyces viscosus* [1] and *Schaalia meyeri* [2]) | | 1 | 7 |
| *Schaalia radingae* (formerly *Actinomyces radingae*) | 8 | 8 | | | | 8 |
| *Schaalia turicensis* (formerly *Actinomyces turicensis*) | 1 | 1 | | 1 (*Gardnerella vaginalis*) | 1 | 3 |
| *Schaalia* sp. | 3 | | 3 (*S. odonotolyltica* [1] and *S. meyeri* [1]) | | 2 | 5 |
| **Streptomyces albidus** | | | | | 1 | 1 |
| **Streptomyces cyaneofuscatus** | | | 1 (*Streptomyces griseus*) | | 1 | 2 |
| **Streptomyces diastaticus** | | | | | 1 | 1 |
| **Streptomyces glauciniger** | | | | | 1 | 1 |
| **Streptomyces thermoviolaceus** | | | | | 3 | 3 |
| *Streptomyces* sp. | 2 | – | 2 (*Streptomyces griseus*) | | 10 | 12 |
| **Thermoactinomyces** sp. | | | | | 1 | 1 |
| *Trueperella bernardiae* | 11 | 11 | | | | 11 |
| *Trueperella pyogenes* | 1 | 1 | | | | 1 |
| **Turicibacter sanguinis** | | | | | 2 | 2 |
| **Williamsia** sp. | | | | | 1 | 1 |
| *Winkia neuii* subsp. *anitratus* (formerly *Actinomyces neuii* subsp. *anitratus*) | 2 | | 2 (*Schaalii neuii/S. radingae*) split | | | 2 |
| *Winkia neuii* subsp. *neuii* (formerly *Actinomyces neuii* subsp. *neuii*) | 2 | 2 | | 1 (*Kytococcus sedenteris*) | | 3 |
| Total | 265/451 (59%)[a] | 170/265 (64.2%)[c] and 170/344 (49.4%)[d] | 71/265 (26.9%)[c] | 40/451 (9%) | 141/451 (31%) | 451 |

[a]All isolates were analyzed by both VITEK MS and 16S rRNA gene sequencing as the reference method. Proteomics and genomic results were compared to phenotypic results (i.e., Gram stain and additional phenotypic testing).

[b]Species not included or limited species representation in the VITEK MS databases: v.2.0 (used 2012–2014), v.3.0 (used 2016–2017) and v.3.2 (2018–2019) are highlighted in bold; none of these specific genera and/or species were in the database in use at the time of proteomics analysis and had not been added as of v.3.2.

[c]Accurate genus and species identification based on the inclusion of the organism in the currently used version of the VITEK MS database. VITEK MS cannot distinguish *L. gasseri* and *L. acidophilus* or *L. casei* and *L. paracasei*, or *Lactiplantibacillus plantarum* and *Lactiplantibacillus paraplanatarum*.

[d]Accurate genus and species identifications based on the total number of clinically relevant isolates studied. The total isolates scored for species-level identification by proteomics was decreased in this column if 16S only provided a genus-level identification. Newer nomenclature has been published for several of these organisms, including *Gleimia europaea, Schaalia meyeri, Schaalia odontolytica* (formerly *Actinomyces odontolyticus*), *Schaalia radingae, Schaalia turicensis, Winkia neuii* subsp. *anitratus*, and *Winkia neuii* subsp. *neuii*.

[e]Based on a multi-sequence alignment using a reference strain and review of the molecular species identification according to the Clinical and Laboratory Standards Institute MM-18 Guidelines (24).

[f]*C. diphtheriae* cannot be distinguished from *Corynebacterium rouxii* using MALDI-TOF, and a longer 16S sequence (~1,060 bp) is required to separate these two species using genomic analysis (25). *C. kroppenstedtii* is closely related to *Corynebacterium parakroppenstedtii* and *Corynebacterium pseudokroppenstedtii* that were more recently taxonomically separated (1), but this change has not been included yet in the instrument's database.

[g]"–", none.

**TABLE 2** Performance of VITEK MS compared to molecular identification of aerobic GPCs[a,e,f]

| Reference method | VITEK MS results | | | | | Total |
|---|---|---|---|---|---|---|
| 16S RNA gene sequencing result[d] | No. (%) of correct identifications at genus level | No. (%) of correct identifications at genus and species levels | No. (%) with discordant species results[b] | No. (%) with wrong result[b] | No. (%) with no results | Total no. (%) |
| *Abiotrophia defectiva* | 1 | 1 | | 1 (*Abiotrophia defectiva/ Paenibacillus* sp.) | | 2 |
| *Aerococcus christensenii* | | | | 1 (*Gardnerella vaginalis*) | 1 | 2 |
| *Aerococcus sanguinicola* | 16 | | 1 (*Aerococcus viridans*) | 1 (*Brevibacillus* sp.) | | 17 |
| *Aerococcus urinae* | 4 | 4 | | | | 4 |
| *Aerococcus vaginalis* | 4 | 4 | | | | 4 |
| *Aerococcus* sp. | 1 | –[g] | 1 (*A. vaginalis*) | | | 1 |
| *Alloiococcus otitis* | 1 | 1 | | | | 1 |
| *Deinococcus wulumuqiensis* | | | | | 1 | 1 |
| *Dolosigranulum pigrum* | 4 | 4 | | 1 (*Facklamia hominis*) 1 (*Paenibacillus* sp.) | 1 | 7 |
| *Enterococcus avium* | 1 | 1 | | | | 1 |
| *Enterococcus pallens* | | | | 1 (*Empedobacter brevis*) | | 1 |
| *Enterococcus faecium* | 1 | 1 | | | | 1 |
| *Enterococcus hirae* | 1 | 1 | | | | 1 |
| *Enterococcus raffinosus* | 2 | 1 | 1 (*Enterococcus raffinosus/E. avium*) | | | 2 |
| *Facklamia hominis* | 1 | 1 | | | | 1 |
| *Facklamia languida* | 1 | | | | 1 | 1 |
| *Facklamia* sp. | 7 | - | 7 (*Facklamia hominis*) | | 1 | 8 |
| *Gemella bergeri* | 2 | 2 | | | | 2 |
| *Gemella haemolysans* | 7 | 7 | | | 1 | 8 |
| *Gemella morbillorum* | 15 | | 6 (*Gemella* sp. splits [5] and *Gemella sanguinis* [1]) | | | 15 |
| *Gemella sanguinus* | 1 | 1 | | | | 1 |
| *Globicatella sulfidifaciens* | 2 | | 1 (*Globicatella sulfidifaciens/ Globicatella sanguinis* split) | 1 (*Gemella sanguinis*) | | 2 |
| *Granulicatella adiacens* | 5 | 5 | | | | 5 |
| *Helcococcus kunzii* | 8 | 8 | | | | 8 |
| *Helcococcus sueciensis* | | | | | 1 | 1 |
| *Kocuria carniphila* | | | | | 1 | 1 |
| *Kocuria rhizophila* | 11 | 11 | | | 1 | 12 |
| *Kocuria* sp. | 1 | - | 1 (*Kocuria rhizophila*) | 1 (*Corynebacterium striatum*) | | 2 |
| *Lactococcus garvieae* | 3 | 3 | | | | 3 |
| *Leuconostoc citreum* | 1 | 1 | | | | 1 |
| *Leuconostoc lactis* | 14 | 14 | | | 1 | 15 |
| *Leuconostoc mesenteroides* | 2 | 2 | | | | 2 |
| *Leuconostoc* sp. | 1 | | 1 (*Leuconostoc* split) | | | 1 |
| *Macrococcus* sp. | | – | | | 1 | 1 |
| *Paracoccus* sp. | | – | | | 3 | 3 |
| *Pediococcus acidilactici* | 2 | 2 | | | | 2 |
| *Pediococcus* sp. | 1 | – | | | | 1 |
| *Planococcus donghaensis* | | | | | 1 | 1 |
| *Staphylococcus argenteus* | 2 | | 2 (*Staphylococcus aureus*) | | | 2 |
| *Staphylococcus arlettae* | 2 | 2 | | | | 2 |
| *Staphylococcus aureus* | 1 | 1 | | | 1 | 2 |
| *Staphylococcus lugdunensis* | | | | | 1 | 1 |
| *Staphylococcus pseudintermedius* | 2 | 2 | | | | 2 |

(*Continued on next page*)

**TABLE 2** Performance of VITEK MS compared to molecular identification of aerobic GPCs[a],[e],[f] (Continued)

| Reference method | VITEK MS results | | | | | Total |
|---|---|---|---|---|---|---|
| 16S RNA gene sequencing result[d] | No. (%) of correct identifications at genus level | No. (%) of correct identifications at genus and species levels | No. (%) with discordant species results[b] | No. (%) with wrong result[b] | No. (%) with no results | Total no. (%) |
| Staphylococcus saccharolyticus | 1 | 1 | 1 (Staphylococcus lugdunensis) | | 1 | 2 |
| Staphylococcus schleiferi | 1 | 1 | | | | 1 |
| Streptococcus agalactiae | | | | | 1 | 1 |
| Streptococcus anginosus | 1 | 1 | | | | 1 |
| Streptococcus cristatus | 1 | 1 | | | | 1 |
| Streptococcus dysgalactiae | 3 | | 2 (S. dysgalactiae/S. equisimilis split) | 1 (Streptococcus pyogenes/S. dysgalactiae/S. gallolyticus split) | | 3 |
| Streptococcus dysgalactiae subsp. dysgalactiae | 1 | 1 | | | | 1 |
| Streptococcus dysgalactiae subsp. equisimilis | 1 | 1 | | | | 1 |
| Streptococcus equinus | 2 | | 1 (Streptococcus lutetiensis/S. infantarius) | | 1 | 2 |
| Streptococcus gallolyticus | 1 | 1 | | | | 1 |
| Streptococcus gallolyticus subsp. macedonicus | 1 | | 1 (Streptococcus gallolyticus subsp. gallolyticus) | | | 1 |
| Streptococcus gallolyticus subpp. pasteurianis | 2 | 2 | | 1 (Paracoccus yeei) | | 3 |
| Streptococcus gordonii | 7 | 7 | | | | 7 |
| Streptococcus infantarius | | | | | 1 | 1 |
| Streptococcus infantis | | | | 1 (Granulicatella adaciens) | | 1 |
| Streptococcus mitis | 5 | | 5 (Streptococcus pneumoniae/ Streptococcus pseudopneumoniae [2], S. mitis/S. oralis split [1], S. pneumoniae [1], and S. parasanguinis [1]) | | | 5 |
| Streptococcus mutans | 2 | 2 | | | 1 | 3 |
| Streptococcus oralis | 2 | | 2 (S. mits/S. oralis [1] and S. parasanguinis [1]) | | | 2 |
| Streptococcus parasanguinis | 1 | 1 | | | | 1 |
| Streptococcus pseudoporcinus | 4 | 3 | 1 (Streptococcus porcinus) | | | 4 |
| Streptococcus salivarius | | | | | 2 | 2 |
| Streptococcus suis | 1 | 1 | | | | 1 |
| Streptococcus thermophilus | | | | 1 (Propionibacterium propionicum) | 1 | 2 |
| Streptococcus urinalis | | | | 1 (Streptococcus agalactiae) | | 1 |
| Streptococcus sp. | 2 | – | 2 (S. pneumoniae) | | | 2 |
| GPC (unidentified)[d] | | | | | 1 | 1 |
| Total | 166/204 (81%)[c] | 103/165 (62.4%)[c] 103/191 (53.9%)[d] | 43/166 (26%)[c] | 14/204 (7%) | 25/204 (12%) | 204 |

[a]All isolates were analyzed by both VITEK MS and 16S rRNA gene sequencing as the reference method. Proteomics and genomic results were compared to phenotypic results (i.e., Gram stain and additional phenotypic testing).

[b]Species not included or limited species representation in the VITEK MS databases: v.2.0 (used 2012–2014), v.3.0 (used 2016–2017), and v.3.2 (2018–2019) are highlighted in bold; none of these specific genera and/or species were in the database in use at the time of proteomics analysis and had not been added as of v.3.2.

[c]Accurate genus and species identification based on the inclusion of the organism in the currently used version of the VITEK MS database.

[d]Accurate genus and species identifications based on the total number of clinically relevant isolates studied. The total number of isolates scored for species-level identification by MALDI-TOF MS decreased in this column if 16S only provided a genus-level identification. One GPC could not be identified by either method.

[e]Based on a multi-sequence alignment using a reference strain and review of the molecular species identification according to the Clinical and Laboratory Standards Institute MM-18 Guidelines (28).

[f]GPC, gram-positive coccus.

[g]"–", none.

identification at genus or species level used interpretive criteria outlined in Clinical and Laboratory Standards Institute (CLSI), Approved Guidelines MM-18 for targeted DNA sequencing analysis (23).

## Data analysis

Data were entered into a Microsoft Excel spreadsheet (MS Office 2016) and analyzed according to standard descriptive methods. VITEK MS performance was calculated against the ability of current instrument databases (i.e., v.2.0, v.3.0, or v.3.2) to accurately identify GPOs at the time of the study. VITEK MS performance was compared to the "gold standard" method (i.e., 16S rRNA gene sequencing). Results from GPOs and GPC identification were analyzed separately according to Tables 1 and 2, respectively. The total number of isolates scored for accurate species-level identification by MALDI-TOF was reduced by the number of isolates for which 16S provided only genus-level identification.

## RESULTS

### Clinical specimens and isolates

Patients' diagnoses included bloodstream infection, pneumonia, lung abscesses, septic arthritis, peritonitis, deep organ and intra-abdominal abscesses, deep wound infections, urinary tract infections, central line infections, prosthetic implant infections, and deep skin and soft tissue infections, including burns. A total of 656 gram-positive bacterial isolates were recovered from 648 specimens; more than a single organism was recovered from only seven (1%) specimens. However, one GPB isolate could not be identified by either MALDI-TOF or 16S sequencing and was not included; 655 GPOs were therefore enrolled, including 451(68.9%) aerobic GPOs and 204 (31.1%) GPCs. This represents between <1% per annum of all invasive isolates identified during the study period. GPOs were mainly recovered from bloodstream infections (35%), sterile fluids and deep tissues (35%), and abscesses/deep wounds (17%) (Fig. 1a). Clinically relevant GPOs were also frequently recovered from patients with synovitis, peritonitis, lower respiratory tract infection, and deep tissue and wound infections. GPCs were mainly recovered from bloodstream infections (53%), but 29% were recovered from deep wounds and other types of specimens. Clinically relevant GPCs were frequently found in urine (i.e., *Aerococcus* spp.) and implant infections (i.e., prosthetic joint, prosthetic heart valve, and central line infections). Enrollment of isolates remained constant with an average of 105 (range = 81–136) isolates accrued each per year (data not shown).

### Distribution and types of recovered gram-positive organisms

The distribution and types of GPBs included are outlined in Table 1. A total of 36 genera and 118 different species were characterized; the largest genera included *Actinomyces* spp. (*n* = 46, 10.3%), *Corynebacterium* spp. (*n* = 89, 20%), *Dermabacter* spp. (*n* = 35, 7.9%), *Schaalia* spp. (*n* = 28, 6.2%), *Rothia* spp. (*n* = 29, 6.4%), and *Bifidobacterium* spp. (*n* = 23, 5.2%), but a diverse spectrum of aerobic GPBs species was represented. The most identified species included *Arcanobacterium haemolyticum*, *Gleimia europaea (*formerly *Actinomyces europaeus*), *Actinomyces israelii*, *Actinomyces naeslundii*, *Lancefieldella parvulus* (formerly *Atopobium parvulum*), *Bifidobacterium breve*, *Corynebacterium kroppenstedtii*, *Corynebacterium tuberculostearicum*, *Dermabacter hominis*, *Gardnerella vaginalis*, *Listeria monocytogenes*, *Rothia aeria*, *Rothia dentocariosa*, and *Trueperella bernardiae* (Table 1). Overall, accurate genus-level identification of GPBs by VITEK MS remained stable over the study as few new genera (i.e., *Janibacter* and *Streptomyces*) were added to the instrument's database. Accurate species-level identification of GPBs by VITEK MS improved over the study, and *Actinomyces*, *Atopobium*, *Bifidobacterium*, and *Corynebacterium* spp. were added to the instrument's database.

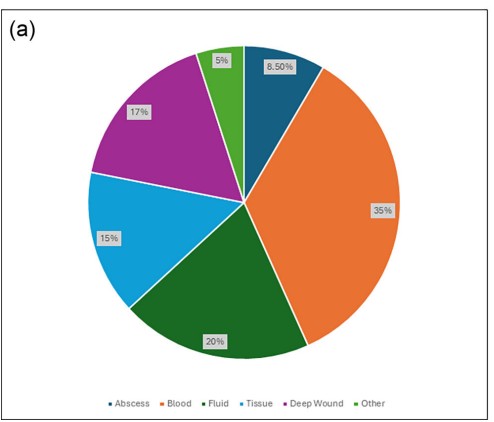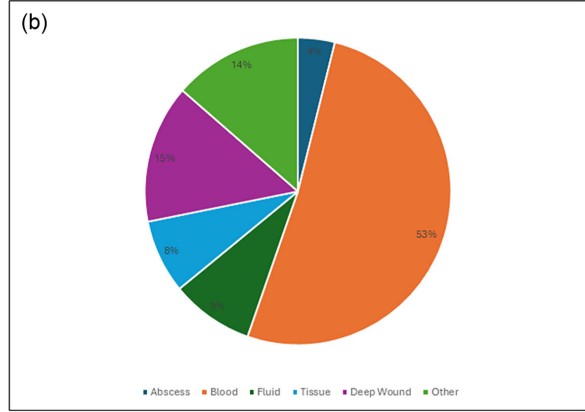

FIG 1 (a) Recovery of gram-positive bacilli from clinical specimens. (b) Recovery of gram-positive cocci from clinical specimens.

The distribution and type of GPCs included are outlined in Table 2. Aerobic GPCs represented 22 different genera and 66 individual species. The most common genera were *Streptococcus* spp. (*n* = 47, 22.7%), *Aerococcus* spp. (*n* = 28, 13.5%), *Gemella* spp. (*n* = 24, 12%), *Lactococcus* (*n* = 17, 8.2%), *Kocuria* spp. (*n* = 15, 7.4%), and *Staphylococcus* spp. (*n* = 12, 5.8%), but a diverse spectrum of organisms was tested. The most identified species included *Aerococcus sanguinicola*, *Gemella morbillorum*, *Kocuria rhizophila*, and *Lactococcus lactis*. Overall, accurate genus-level identification of GPCs by VITEK MS remained stable over the study. Accurate species-level identification of GPCs by VITEK MS improved over the study with the addition of *Aerococcus*, *Enterococcus*, *Kocuria*, *Leuconostoc*, *Planococcus*, *Staphylococcus*, *and Streptococcus* spp. to the instrument's database.

## Performance of VITEK MS compared to 16S rRNA gene sequencing

Overall, 16S sequencing accurately identified 99.8% of GPOs at the genus level and 82.1% at the species level. In comparison, VITEK MS accurately identified 65.6% and 41.7% of GPOs at the genus and species levels, respectively. Approximately one-third (26.3%) of GPO isolates accurately identified at the genus level by MALDI-TOF had a discrepant species result. Wrong identifications and no results occurred for 8.4% and 26.0% of GPO isolates, respectively. No results occurred for a third of all GPBs but only 13% of GPCs.

16S accurately identified 99.8% GPB isolates at the genus level and 77% at the species level. One GPB could not be identified by either MALDI-TOF or 16S analyses (not included in Table 1). Performance of VITEK MS for the identification of GPBs is shown in Table 1. MALDI-TOF accurately identified 9.2% and 49.4% of isolates at the genus and species levels, respectively. Discordant species-level identifications are also shown in Table 1. No result (31%) or a wrong identity (9%) occurred for 13% of GPB because organisms were not included in VITEK MS databases (highlighted in bold in Table 1). Wrong identifications occurred for *Gleimia europaea*, *Actinomyces gerencseriae*, *Actinomyces graevenitzii*, *Winkia neuii* subsp. *neuii* (formerly *Actinomyces neuii* subsp. *neuii*), *Schaalia turicensis* (formerly *Actinomyces turicensis*), *Actinomyces* spp., *Lancefieldella parvula* (formerly *A. parvulum*), *Fannyhessea vaginae* (formerly *Atopobium vaginae*), *Atopobium* sp., *Arthrobacter* sp., *Bacillus thuringiensis*, *Brevibacterium paucivorans*, *Corynebacterium urealyticum*, *Lactocaseibacillus animalis* (formerly *Lactobacillus animalis*), *Lactobacillus crisptatus*, *Lactobacillus gasseri*, *Lactobacillus iners*, *Limosilactobacellus vaginalis* (formerly *Lactobacillus vaginalis*), *Microbacterium aurum*, *Microbacterium lacticum*, *Microbacterium* sp., *Paenibacillus* sp., *Renibacterium* sp., *Rhodococcus equi*, *Rhodococcus* sp., *Rothia dentocariosa*, and *Rothia* sp. (Table 1).

16S accurately identified 99.5% GPC isolates at the genus level and 92.3% at the species level. Performance of VITEK MS for the identification of GPCs is shown in Table 2. MALDI-TOF accurately identified 81% and 53.9% isolates at the genus and

species levels, respectively. Discordant species-level identifications are shown in Table 2. No results (12%) or a wrong identity (7%) occurred for 5.3% of GPCs because organisms were not included in the VITEK MS database (highlighted in bold in Table 2). Wrong identifications occurred for *Abiotrophia defectiva*, *Aerococcus christensenii*, *A. sanguinicola*, *Bacterium* sp., *Dolosigranulum pigrum*, *Globicatella sulfidifaciens*, *Rothia koreensis* (formerly *Kocuria koreensis*), *Streptococcus infantis*, *Streptococcus gallolyticus* subsp. *pasteurianus*, and *Streptococcus thermophilus* (Table 2). One GPC isolate could not be identified by either 16S sequencing or MALDI-TOF MS.

## DISCUSSION

This study uniquely highlights the utility and pitfalls of MALDI-TOF for accurate identification of rare and unusual GPOs within a sequential clinical microbiology workflow. Although most clinically encountered GPOs were accurately identified during the multi-year study period (data not shown) by 16S sequencing, MALDI-TOF MS performance for study organisms was less optimal. VITEK MS gave an accurate genus result for 59.2% of GPB and 81% of GPC isolates but gave a wrong identification or no results for 9% of GPB vs 7% of GPC and 31% of GPB vs 12% of GPC isolates, respectively. VITEK MS has difficulty identifying approximately 5%–13% of GPOs due to limitations of the instrument's database over the study period, although the addition of several key genera and species to the v.3.0 and v.3.2 databases improved identification of some GPB genera and GPC species in the later part of the study. 16S provided an accurate genus identification for all but one GPB isolate, but species identification could improve with sequencing of a much larger portion of the 16S rRNA gene up to ~1,160 bp; many GPOs have an almost complete homology within the 16S V1–V3 gene regions (approximately the first 500 bp interrogated by the fast MicroSEQ 500 16S DNA PCR kits was used) (23).

Prior studies focused on the use of MALDI-TOF MS for the identification of clinically relevant bacteria in the clinical laboratory have mainly included gram-negative bacteria and gram-positive cocci and included only a few GPOs (11, 14, 24, 26–29). Limited prior studies have evaluated the VITEK MS system for GPB identification, nor have any studies compared the performance of Bruker to the VITEK MS (2, 15). Navas and colleagues evaluated the accuracy of VITEK MS identification for two to six aerobic gram-positive bacilli representing 20 genera and 38 species and found correct identifications for 85% of the isolates (i.e., 28.6% at the genus level, 56.3% at the species level), while 7.3% were misidentified (30). Individual reports of MALDI-TOF MS analysis for a single GPO genera have mainly included *Corynebacterium* spp. (17, 18). *Actinomyces* spp. (12, 19, 30), *Listeria* spp. (20), and difficult to identify GPOs (29). Schulthess and colleagues used the Bruker MALDI Biotyper and compared the effects on the performance of the three methods for sample preparation for the identification of 190 well-characterized GPOs (15). Rates of identification were improved using a direct-transfer formic acid extraction preparation and a species cutoff of 1.7. A total of 215 clinical GPB isolates were then studied using these methods, and the results were compared to conventional methods with discrepancies being resolved by 16S rRNA and *rpo*B gene sequencing analysis. Identification accuracy at the genus and species levels of 87.4% and 79.1%, respectively, was achieved. The rate of non-identified isolates also decreased from 12.1% to 5.6% using the Bruker database amended by reference spectra of the 190 GPOs studied (15). Barberis and colleagues also compared the Bruker system and conventional phenotypic methods to identify 333 GPO clinical isolates comprising 22 genera and 60 species; 16S rRNA sequencing was the reference molecular method to resolve discrepancies, and *rpo*B gene sequencing was secondarily done on *Corynebacterium* spp. that could not be identified (2). Bruker had higher efficiency for identification of *Corynebacterium* spp. (92%) and *Actinomyces* spp. (89%) and related genera but was less efficient for pigmented GPOs (i.e., *Leisonia* spp., *Microbacterium* spp., *Exiguobacterium* spp., *Cellulosimicrobium* spp., and *Brevibacterium/Arthrobacter* spp.) (2).

Schulthess and colleagues also used the Bruker MALDI Biotyper and compared the effects on performance of the three methods for sample preparation for the

identification of 156 well-characterized GPCs (16). Rates of identification at the genus level were approximately 99% regardless of the sample preparation/extraction process used. However, the species identification rate was higher (77.6%) using the direct-transfer formic acid method (16). A subsequent prospective study of 1,619 clinical GPC isolates compared to conventional phenotypic methods showed a congruence of 95.6% for MALDI-TOF identification (16). Limitations of the Burker system for the identification of GPCs included differentiation of members of the *Streptococcus mitis* group and identification of *Streptococcus dysgalactiae*. Lee and colleagues performed a comparative evaluation of the VITEK MS and Bruker MALDI-TOF systems for the identification of GPCs routinely isolated in the clinical laboratory, including *Staphylococcus* spp., beta-hemolytic and viridans *Streptococcus* spp., and *Enterococcus* spp. (14). Both MALDI-TOF systems accurately identified 97.2% and 94.7% of 394 isolates; Bruker had a lower accuracy for the identification of *S. mitis* group isolates, particularly *S. pneumoniae*, which has been reported by others (14, 28). Overall concordance rates between the two MALDI-TOF MS systems were 91.9% at the species level, while the concordance rates for staphylococci and streptococci were 92.1% and 88.1%, respectively (14).

Approximately 5%–13% of proteomic errors in our study occurred because specific organisms were not included in VITEK MS databases (Tables 1 and 2 highlighted in bold). However, MALDI-TOF errors also occurred for many organisms whose spectral profiles were available in the instrument's database at the time of testing despite repeated analyses. These types of MALDI-TOF MS errors have been reported to occur because a quality peak is difficult to distinguish with specific spectra (31). MALDI-TOF MS may also have difficulty separating some genera and species, giving lower identification scores related to interspecies similarities (22, 23). Many important GPOs also have closely related 16S genetic sequences and MALDI-TOF spectral profiles, decreasing these methods' ability to give an accurate identification (29). The limitations and pitfalls of 16S sequencing performance and interpretation have been outlined in detail in published CLSI guidelines (23) and in the review by Church and colleagues (32). Many GPOs, particularly GPBs, require a longer 16S sequence up to ~1,060 bp to be identified at the species level (23, 33). Additional gene targets such as the *rpo*B gene may also enhance molecular identification of GP (23, 33). 16S gene sequencing is a valuable complementary method to proteomics, and it should remain in place within our laboratory's workflow until sufficient proteomic spectra for rare and unusual GPOs are included in VITEK MS instrument databases.

A limitation of this work includes the enrollment of isolates from a single Canadian center, although our laboratory performs all microbiological testing for the entire regional population. Many smaller clinical laboratories may therefore not encounter many of these rare or unusual GPOs on the same frequency as our large, centralized reference laboratory. Geographic strain differences for GPOs may cause discordant identification rates, and limited overall data comparing MALDI-TOF results across MALDI-TOF MS platforms have been globally reported. Chart reviews may have improved assessment of pathogenicity of some isolates, but each case was medically assessed prior to enrollment. Although only genus identity may suffice for patient care, rare and unusual GPO species have unique inherent or acquired multi-drug resistance profiles and epidemiologic patterns of disease necessitating full speciation. Laboratories should have a workflow for the identification of pathogenic unusual or rarely encountered aerobic GPOs that includes 16S rRNA gene sequencing whenever MALDI-TOF cannot give a definitive identification. Ongoing studies are needed to document the performance of MALDI-TOF MS systems used in the clinical laboratory for accurate identification of clinically relevant gram-positive bacteria.

## ACKNOWLEDGMENTS

We thank the medical laboratory technologists at Alberta Precision Laboratories for their assistance with isolate testing.

This study was unsupported.

D.L.C.: formal analysis, conceptualization, data curation, methodology, project administration, and writing (original draft, review, and editing); T.G. and D.G.: methodology and writing (review and editing).

## AUTHOR AFFILIATIONS

[1]Department of Pathology and Laboratory Medicine, Cummings School of Medicine, University of Calgary, Calgary, Alberta, Canada

[2]Department of Medicine, Cummings School of Medicine, University of Calgary, Calgary, Alberta, Canada

[3]Alberta Precision Laboratories (formerly Calgary Laboratory Services), Calgary, Alberta, Canada

## AUTHOR ORCIDs

D.L. Church  http://orcid.org/0009-0008-9941-9050

## AUTHOR CONTRIBUTIONS

D.L. Church, Conceptualization, Formal analysis, Methodology, Writing – review and editing | T. Griener, Conceptualization, Methodology, Writing – review and editing | D. Gregson, Conceptualization, Formal analysis, Investigation, Methodology, Writing – review and editing

## DATA AVAILABILITY

The data that support the findings of this study are available from Alberta Health Services (AHS), Alberta Precision Laboratories (APL) (formerly Calgary Laboratory Services), but restrictions apply to the availability of these data, which have been used under the ethics agreement for the current study and so are not publicly available. However, data are available from the authors upon reasonable request and with permission from AHS/APL.

## ADDITIONAL FILES

The following material is available online.

Open Peer Review

**PEER REVIEW HISTORY (review-history.pdf).** An accounting of the reviewer comments and feedback.

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
