## [Reviewer comments · Microbiology Spectrum]

Microbiology Spectrum

Multi-Year Comparison of VITEK® MS performance for identification of rarely encountered pathogenic gram-positive organisms (GPOs) in a large integrated Canadian healthcare region.

Deirdre Church, Thomas Griener, and Dan Gregson

Corresponding Author(s): Deirdre Church, University of Calgary Cumming School of Medicine

Review Timeline:

Submission Date:	October 8, 2024
Editorial Decision:	January 12, 2025
Revision Received:	February 13, 2025
Editorial Decision:	February 24, 2025
Revision Received:	February 26, 2025
Accepted:	March 22, 2025

Editor: Rebecca Yee

Reviewer(s): The reviewers have opted to remain anonymous.

Transaction Report:

DOI: <https://doi.org/10.1128/spectrum.02545-24>

Re: Spectrum02545-24 (Multi-Year Comparison of VITEK® MS performance for identification of rarely encountered pathogenic gram-positive organisms (GPOs) in a large integrated Canadian healthcare region.)

Dear Dr. Deirdre L. Church:

Thank you for the privilege of reviewing your work. Below you will find instructions from the Spectrum editorial office, and the reviewer comments.

Revision Guidelines

Sincerely,
Rebecca Yee
Editor
Microbiology Spectrum

Reviewer #1 (Comments for the Author):

It could be argued that MALDI identifications that yield slashline results (e.g. *S. mitis*/*S. oralis*) should be considered correct, especially for instances where the individual identifications are not the claimed identification in the package insert of the database - *S. mitis*/*S. oralis* is an example where the approved display is the slashline result. This may also apply to "non claimed" but displayed identifications as well.

Were organisms repeated if they gave slashline MALDI result?

Did all isolates have a 16s identity score of 99.9 to be included (line 146)?

Reviewer #3 (Comments for the Author):

Major Comments

The authors addresses identification using Vitek MS compared with 16S as gold standard.

- line 126-127. In spite of this technology being used for more than 10 years the authors should describe what Biomerieux defines as: ID with high (unambiguous) confidence, low confidence or absent result.

The authors should also describe if they are only comparing spectra towards the Biomerieux FDA "cleared" database (with respect to results) or if they included any results from testing against a research library.

- Line 130 low and high confidences are sort of defined but sentence content could be better described. The authors should also specifically address how IDs were ascertained if strains had high MALDI scores (>99%) towards two or more taxa; such organisms generally can not be differentiated by 16S alone either. This would have occurred for example with *B. cereus* and *B. thuringiensis*.

Line 146, definition of definitive ID by 16S; the authors should describe what is meant (in the tables, text) where ID is noted as "Carnobacterium sp", that is, was the 16S < 99.0% to 1 or more sequences from type strains from that genus? Other?

- Line 146-147, the authors should describe why some strains are described only as for example *Janibacter* sp.; was this because by Smartgene the sequence was < 99% to a single species in the genus? < 99% towards multiple species in genus? It would be useful to describe briefly here (from CLSI ref?).

- Similarly from Table 1, what criteria was used to assign ID only to a family level name (*Thermoactinomycetaceae*)

- From table 2, the authors ID two strains as *Paracoccus pancisoli*/ *Paracoccus sanguinis*. The reason for this wording should be explained along with other definitions (above)

- Text, footnotes to figures: Authors should briefly describe the phenotypic workup is done to support ID of common GPOs or other taxa described here, as many labs no longer have ready access to a wide variety of biochemical test assays. Authors should modify wording in footnotes from phenotypic "identification" to "additional phenotypic testing" which may not necessarily provide definitive ID.

- Authors should better clarify the meaning of 'wrong ID', where significant discrepancy had occurred between the two methods.

- The authors stopped this study in 2019, but it appears that ver 3.2 of MS library is/may also be the current one (in late 2024). The authors should comment on critical nature of this possible lack of updates of new data and modification of nomenclature as changes are proposed, over a 4 year period.

Lines 188-194, Table content: this journal will insist that the current validly named Latin binomial be used for all taxa; CAP and other organizations may require this information if name impacts on treatment options.

- the authors should comment if both Smartgene.com website and Biomerieux are using the old names as shown in places here or if the authors are choosing to use the older names being in mind that Genbank for example makes a concerted effort to use newer names. Suggest using the List of Prokaryotes with Standing in Nomenclature (LPSN, lpsn.dsmz.de) . It is understood that the older names would have come up for older parts of the study but it would be useful for clarity to explain if older name came up in original study but updated nomenclature exists, that the latter should be used in results/discussion. This would eliminate, as an example, *Actinomyces odontolyticus* (shown in Table 1) and *Schaalia odontolytica* (Table 2 ?for some reason) data should be amalgamated.

As well, in the text, if for clarity the authors include both older and newer names, the usual format would be: *Gleimia europaea* (formerly *Actinomyces europeus*). Although this may not be a totally comprehensive list but corrections to the many taxonomy errors or use of older names from the two tables are listed below:

- Table 1: *Gleimia europaea*; *Schaalia georgiae* (also missing from database using either latin name); *Schaalia meyeri*; *Winkia neuii* subsp *neuui* and subsp *anitratus*; *Schaalia odontolyticus*, *S. radingae*, *S. turicensis*.

- what is the definition of the Actinomycetaceae grouping? ID only at genus level for genera not *Actinomyces*, *Schaalia*, *Winkia*?

- *Lancefieldella parvula* should be written out, *A parvulum* removed; modify: *L. rimae*, *Fannyhessa vaginae*;

- *Pseudoglutamicibacter cumminsii*; *Paenarthrobacter nicotinovorans*;

- Authors should describe what definition of *Arthrobacter* sp and sp. LM3 are

- "*Bacillus aerophilus*" is not a validly described species and so should be shown without italics, in quotes (LPSN)

- *Niallia circulans*, *N. nealsonni*, *Peribacillus simplex*,

- Descriptor of ARUPunID 142

- Authors should review *Brevibacillus paucivorans* (does not exist). Do you mean *Brevibacterium paucivorans* (which does exist)

- Authors, FYI, *C. diphtheriae*, by both 16S and MALDI Bruker, can not be readily discerned from newish species *C. rouxii* and answer comes up always as *C. diphtheriae*

- *C. fastidiosum*. This is not a valid bacterial name in LSPN and usage appears to be linked solely to match to a 20y old GB deposit X84245. Should be modified to *Corynebacterium* sp or removed. This would explain why there is no correlate in Vitek MS
- *C. kroppenstedtii*, authors should be aware that some taxa closest to this species have been assigned to *C. parakroppenstedtii* or *C. pseudokroppenstedtii* in 2022. Authors should re-evaluate data to see if their 11 isolates by 16S are correctly identified
- *C. segmentosum* is not validly named in LSPN; should be modified or removed
- Authors should verify: *Dermabacter hominis* is validly named and is not an uncommon isolate; *Dermatobacter hominis* is NOT validly named (as yet). Suspect authors mean the former taxon.
- Similarly, *Dermabacter jinuensis* is validly named but *Dermatobacter jinuensis* (shown) does not exist.
- relevant *Lactobacillus* species should be shown here as assigned by Zhong et al 2020 ISJEM 70:4050-4060 and as shown in LPSN: e.g. *L. animalis* has been reassigned to *Ligilactobacillus animalis*
- *Oceanibacillus massiliensis* is not a valid bacterial name since 2012 and so should be shown without italics and this caveat noted
- *Reinibacterium* genus is not found in the LSPN but *Renibacterium* is a genus in the Micrococcaceae. ; authors should correct or remove
- *Streptomyces rutgersensis* is shown as being a later heterotypic synonym of *Streptomyces diastaticus* in the LPSN. Should be reviewed/corrected.

Table 2.

- According to the LSPN, *Aerococcus vaginae* as shown in several places here does not exist; *Aerococcus vaginalis* is validly named, as is *Anaerococcus vaginalis*. Authors should clarify which taxon is being referred to here
- authors should describe what "Bacterium sp" refers to
- most taxa in these two tables are shown in alphabetical order; as such why is *E. pallens* shown before *E. avium*?
- *Kocuria koreensis* was reassigned to *Rothia koreensis* in 2018 (LPSN)
- *Leuconostoc garlicum* is not validly named; should be corrected to *Leuconostoc* sp or removed (LPSN)
- *Streptococcus bovis* group, a time-honoured grouping, does not exist in LSPN, nor does *S. bovis* (LPSN suggests that it is a synonym of *S. equinus*)
- LPSN suggests that *S. macedonicus* was reassigned to *S. gallolyticus* subsp *macedonicus* in 2003. Data suggests that the subspecies can not be discerned by methods here.
- *S. pasteurianis*, LPSN suggests that this is also a subspecies named *S. gallolyticus* subsp *pasteurianis* change occurred in 2003.

Minor Comments

- suggest describing acronym GPO in first mention in the abstract
- line 67, ref 1, is an old version of an annual review of taxa. Suggest updating.
- line 75-76, indeed, taxa which are biochemically inert make phenotypic testing alone suboptimal but also suggest adding: as well, many species which are reactive have identical or nearly identical phenotypes, that is phenotypic reactions alone may not provide definitive differentiation
- Authors should comment how often the database at Smartgene.com is updated (presumably very frequently)

Reviewer #4 (Comments for the author):

Major comments:

1. In all the your tables where the 'incorrect or discordant' organisms were detected, are these all high scores or low scores (confidence level). Please elaborate in the text.
2. This study was performed on the VITEK but in the Discussion, there appears to be a heavy focus on the limitations of the Bruker system. Are there such similar studies or even case reports on the inaccuracies of the VITEK for similar organisms that you showed to be an issue on the VITEK?
3. Given that the reference method here is sequencing, please address any limitations of the sequencing method as well. There are known limitations and organisms that 16S also fails to identify correctly to the species level.

Minor Comments:

1. Please spell out GPO before using the abbreviation in the abstract
2. Line 68- 'particularly patients with immune compromise,' is awkward. Please revise.
3. "Laboratory setting" in the Methods may be removed as this information is not relevant to how the project was executed. This information regarding the hospital's demographics can be important in the Discussion if it were to describe the diversity of isolates seen at this hospital.
4. This sentence (line 114-116) appears to be missing a word to connect all the thoughts: Subculture of positive blood cultures inoculated blood agar (BA), chocolate agar (CHOC), MacConkey agar (MAC) and Brucella blood agar (BBA) (Dalynn, Calgary) incubated for 5d at 35o C.
5. There is switch between MALDI-TOF and 'proteomics' throughout the manuscript. Please stick to one for clarity.
6. Please ensure all genus/species are italicized throughout the manuscript and tables/figures. Ensure spacing is also accurate. In instances where there are changes to updated nomenclature, please include in parentheses the new name. In essence, both

names should be included for thoroughness, allowing all laboratorians whether they use old or new nomenclature to appreciate this work.

7. Line 220- missing a (in one of the %s

Spectrum02545-24 (Multi-Year Comparison of VITEK MS performance for identification of rarely encountered pathogenic gram-positive organisms (GPOs) in a large integrated Canadian healthcare region.

Response to the Reviewers

Reviewer	Issue	Response	Comments
#1	MS slashline results (e.g. S. mitis/oralis) should be considered correct	Data analysis was reviewed and compared to the appropriate VITEK MS database; results were corrected in Tables 1 and 2 accordingly	All MS slashline results where the VITEK MS databases wouldn't allow separation of species have been review and corrected in the Tables/data analyses
	Were organisms with MS slashline results repeated?	Methods section has been revised	Yes
	Did all isolates have a 16S identify score of 99.9 to be included?	Methods section has been revised	Yes
#3	Line 126-127; describe the parameters for VITEK MS results	Methods section has been revised	A statement has been included to clarify these distinctions according to the manufacturer in the Methods section
	Describe whether the FDA cleared database or research library was used	Methods section has been revised	A statement has been added to the Methods section; only the FDA-cleared databases were used
	Line 130; better clarify how IDs were ascertained if strains had high MALDI Scores (>99%) for two or more taxa	Methods section has been revised	This has been clarified by including a statement in the Methods section
	Line 146; Clarify how 16S sequence analysis was done	Methods section has been revised	Detailed performance and application criteria and results interpretation for 16S sequencing analyses have been previously described by CLSI guidelines (reference 24) and in our CMR review that is now included as reference 33.
	Table 1 Thermoactinomycetaceae ?is this correct	Table 1 has been revised	This is an error; should have been Thermoactinomyces sp.
	Table 2 Paracoccus pansicoli/Paracoccus sanguinis – please clarify	This has been merged with Paracoccus sp. and the split ID deleted	Both methods cannot separate these species in the case of these 2 isolates.
	Table footnotes; change to additional phenotypic testing	Tables 1 and 2 footnotes have been revised	

	Clarify what is a “wrong ID” in the text, footnotes and Tables	This has been defined in the Methods Section and added to the Table footnotes	Wrong ID = inaccurate genus AND species
	Clarify whether the VITEK MS library (databases) is regularly updated to reflect current taxonomic changes	A statement has been added to the Methods and Discussion sections	No, as evident in the release dates for VITEK MS database version releases, they are not frequent enough to update the rate of taxonomic changes to organisms in the bacterial database. We are still using V3.2 currently which was released in 2018 so the manufacturers databases are not helping reassign new names. This places the onus on clinical laboratories to continuously update to new names. This also requires a lot of reworks of LIS databases and comments and education of physicians about taxonomic changes. So Taxonomic changes create lots of work for clinical laboratories.
	Line 188-194 and Tables 1 and 2; update organism names to reflect the most current names (even though what is reflected herein was the actual “result” given by both methods at the time of study.	Extensive changes have been made to the text and Tables 1 and 2 to address all the taxonomic concerns.	Since the changes required to Table 1 resulted in a complete “red-out” of the previously submitted version, the marked-up version with all the taxonomic changes (not highlighted) is submitted as the revised one. Renaming many organisms in Table 1 required an extensive resorting (to be in alphabetical order) and recalculation of some data (e.g. separation of Actinomyces from Schaalialia and Winkia). The new taxonomic name is now the name used with (formerly named) in both Tables.
	Typos in Table 1	Some names listed as not being in the LSPN are due to small typos to the names (i.e., Reinibacterium instead of Renibacterium)	All organism names have been reviewed to ensure they are correctly spelled.
	Concerns about separation of C. diphtheriae/C. rouxii and C. kroppenstedii into the newly assigned	A new footnote (f) has been added to Table 1 outlining these concerns.	C. kroppenstedii would have been the designated name for these organisms during the study. We have not re-blasted

	species (2022).		the 11 sequences to see if the names should be updated.
	Define GPO in the Abstract	This has been done	
	Update Reference 1	This has been done	
	Line 75-76; clarify phenotypic identification	This statement has been revised	
	How often is the Smartgene database revised?	The Methods section has been revised	The bacterial database and centroids database is continually updated – a statement was added.
#4	Define incorrect or discordant results	The Methods section has been revised	Incorrect ID is wrong genus and species results. This discordant results column outlines the differences in species-level identification called by MS when there is genus-level agreement.
	Modify the Discussion to include more VITEK MS reports	A new literature search was performed. Only 1 additional VITEK MS evaluation from 2014 for GPBs was found and it has been included in the Discussion and as a new reference (32).	
	Describe the limitations of 16S sequencing	A new statement has been included in the Discussion referencing the CLSI guidelines (reference 24). Our recent comprehensive CMR review has also been cited (new reference 33).	A full description of the performance of 16S by clinical microbiology laboratories including the limitations and pitfalls is contained in our CMR review.
	Minor comments – address typos and grammatical errors as listed	All the concerns have been addressed through corrections to the manuscript	

Re: Spectrum02545-24R1 (Multi-Year Comparison of VITEK® MS performance for identification of rarely encountered pathogenic gram-positive organisms (GPOs) in a large integrated Canadian healthcare region.)

Dear Dr. Deirdre L. Church:

Thank you for the privilege of reviewing your work. Below you will find my comments, instructions from the Spectrum editorial office, and the reviewer comments.

We thank the authors for their attempt in revising the manuscript and addressing each reviewers' comments. According to your 'Response to Reviewer's Comments' document, the authors made extensive changes to the Methods and Discussion sections. However, in the marked up manuscript, minimal changes were highlighted. Given the extensive changes that were recommended by the reviewers, please include in your point-by-point response which lines numbers have been changed or, alternatively, ensure that the marked-up revision of the manuscript includes all the changes highlighted (in tracked changes). Additionally, the marked-up file included comments for what appears to be for the authors (e.g. 'Check all names', 'Check data') which are distractions for the reviewers. Please re-upload a modified marked-up version of the manuscript that allows the reviewers to clearly see all the changes that were made.

If you cannot complete the modification within 60 days, please contact me. If you do not wish to modify the manuscript and prefer to submit it to another journal, notify me immediately so that the manuscript may be formally withdrawn from consideration by Spectrum.

Revision Guidelines

Sincerely,
Rebecca Yee
Editor
Microbiology Spectrum

Spectrum02545-24 (Multi-Year Comparison of VITEK MS performance for identification of rarely encountered pathogenic gram-positive organisms (GPOs) in a large integrated Canadian healthcare region.

Response to the Reviewers – NOTE: all responses outlined by page/lines refer to the MARKER-UP version of the REVISED document.

Reviewer	Issue	Response	Comments
#1	MS slashline results (e.g. S. mitis/oralis) should be considered correct	Data analysis was reviewed and compared to the appropriate VITEK MS database; results were corrected in Tables 1 and 2 accordingly	All MS slashline results where the VITEK MS databases wouldn't allow separation of species have been review and corrected in the Tables/data analyses
	Were organisms with MS slashline results repeated?	Methods section has been revised; See page 7 lines 138-141	Yes
	Did all isolates have a 16S identify score of 99.9 to be included?	Methods section has been revised; See page 7 Lines 141-147 and page 9, Lines 164-166	Yes
#3	Line 126-127; describe the parameters for VITEK MS results	Methods section has been revised; See page 7 Lines 141-147	A statement has been included to clarify these distinctions according to the manufacturer in the Methods section
	Describe whether the FDA cleared database or research library was used	Methods section has been revised; See page 7 Lines 146-147	A statement has been added to the Methods section; only the FDA-cleared databases were used
	Line 130; better clarify how IDs were ascertained if strains had high MALDI Scores (>99%) for two or more taxa	Methods section has been revised; See page 7 Lines 137-147	This has been clarified by including a statement in the Methods section
	Line 146; Clarify how 16S sequence analysis was done	Methods section has been revised; See page 8, Lines 159-168	Detailed performance and application criteria and results interpretation for 16S sequencing analyses have been previously described by CLSI guidelines (reference 24) and in our CMR review that is now included as reference 33.
	Table 1 Thermoactinomycetaceae ?is this correct	Table 1 has been revised	This is an error; should have been Thermoactinomyces sp.
	Table 2 Paracoccus pansicoli/Paracoccus sanguinis – please clarify	This has been merged with Paracoccus sp. and the split ID deleted	Both methods cannot separate these species in the case of these 2 isolates.
	Table footnotes; change to additional phenotypic testing	Tables 1 and 2 footnotes have been revised to the recommended statement	

	Clarify what is a “wrong ID” in the text, footnotes and Tables	This has been defined in the Methods Section and added to the Table footnotes; See page 7, lines 144-146	Wrong ID = inaccurate genus AND species
	Clarify whether the VITEK MS library (databases) is regularly updated to reflect current taxonomic changes	A statement has been added to the Methods and Discussion sections – See Page7, Lines 146-47	No, as evident in the release dates for VITEK MS database version releases, they are not frequent enough to update the rate of taxonomic changes to organisms in the bacterial database. We are still using V3.2 currently which was released in 2018 so the manufacturers databases are not helping reassign new names. This places the onus on clinical laboratories to continuously update to new names. This also requires a lot of reworks of LIS databases and comments and education of physicians about taxonomic changes. So Taxonomic changes create lots of work for clinical laboratories.
	Line 188-194 and Tables 1 and 2; update organism names to reflect the most current names (even though what is reflected herein was the actual “result” given by both methods at the time of study.	Extensive changes have been made to the text and Tables 1 and 2 to address all the taxonomic concerns. See Page 10-11, Lines 205-214; page 12, lines 244-252; page 13, lines258-262 – all names updated to most recent taxonomic ones as follows: TABLE 1  1. Gleimia europaea (formerly A. europeus) 2. Schaalia meyeri (formerly A. meyeri) 3. Schaalii odontolyticus (formerly A. odonotolyticus) 4. Schaalia radingae(formerly A. radingae) 5. Schaalia turicensis(formerly A. turicensis). 	Since the changes required to Table 1 resulted in a complete “red-out” of the previously submitted version, the marked-up version with all the taxonomic changes (not highlighted) is submitted as the revised one. All totals (%) in the Tables have been updated as has the text to reflect the movement of some organisms from Table 2 to Table 1 (N=3); See page 2 lines 34-39; page 10, lines 191-193; page 10, line 207;page 11, lines 221-222. Page 12 line 242; page 12, lines 255-56; page 14, lines 270-271. Renaming many organisms in Table 1 required an extensive resorting (to be in alphabetical order) and recalculation of some data (e.g. separation of Schaalia and Winkia from Actinomyces). The new taxonomic name

		 6. Winkia neuii subsp. Neuui(formerly A. neuii subsp. Neuui) 7. Acintinomycetaceae grouping merged with Actinomyces spp. 8. Lancefieldella parvula (formerly A. parvulum) 9. Lanfieldella rimae(formerly A. rimae) 10. Fannyhessea vaginae (formerly A. vaginae) 11. Pseudoglutamicibacter cumminsii 12. Paenarthrobacter nicotinovorans 13. Arthrobacter spp LM3 merged with Arthrobacter spp. 14. Bacillus aerophilus has been unitalicized 15. Niallia circulans 16. Niallia nealsonni 17. Peribacillus simplex 18. ARUPunID142 has been deleted 19. Brevibacillus paucivorans has been corrected to Brevibacterium paucivorans 20. C. fastidiosum has been included in Corynebacterium sp. 21. C. segmentosum has been merged with Corynebacterium sp. 22. Dermatobacter hominis and Dermatobacter jinjuensis changed to Dermatobacter 23. Lactobacillus species were reviewed and their names updated to the most recent taxonomy using the (formerly genus-species) format 	is now the name used with (formerly named) in both Tables.
--	--	--	---

		24. Oceanibacillus massiliensis has been unitalicized 25. Streptomyces rutgersensis has been changed to S. diastaticus TABLE 2  1. Aerococcus vaginae has been changed to A. vaginalis 2. Bacterium sp. has been removed as a valid identification and added to the bottom as a GPC that was unidentified by both methods; See page 12 line 253 for recalculation of 16S results 3. E. avium has been moved to be listed before E. pallens 4. Rothia koreensis (formerly Kocuria koreensis) – moved to Table 1 5. Leuconostoc garlicum removed and added to Leuconostoc sp 6. S. bovis group removed and added to S. equinis 7. S. macedonicus left under but included as a subsp. S. gallolyticus 8. S. pasteurianis moved to S. gallolyticus subsp pasteurianis 	
	Typos in Table 1 and 2	Some names listed as not being in the LPSN are due to small typos to the names (i.e., Renibacterium)	All organism names have been reviewed against LPSN to ensure they are correctly spelled.
	Concerns about separation of C. diphtheriae / C. rouxii and C. kroppenstedii into the newly assigned species (2022).	A new footnote (f) has been added to Table 1 outlining these concerns.	C. kroppenstedii would have been the designated name for these organisms during the study. We have not re-blasted the 11 sequences to see if the names should be updated.
	Define GPO in the Abstract	This has been done	

	Update Reference 1	This has been done	
	Line 75-76; clarify phenotypic identification	This statement has been revised; See page 7 Lines 129-131	
	How often is the Smartgene database revised?	The Methods section has been revised; See page 8, Lines 163-164	The bacterial database and centroids database is continually updated – a statement was added.
#4	Define incorrect or discordant results	The Methods section has been revised; See page 7, Lines 141-146	Incorrect ID is wrong genus and species results. This discordant results column outlines the differences in species-level identification called by MS when there is genus-level agreement.
	Modify the Discussion to include more VITEK MS reports	A new literature search was performed. Only 1 additional VITEK MS evaluation from 2014 for GPBs was found and it has been included in the Discussion and as a new reference (32). See page 14, Lines 283-286.	
	Describe the limitations of 16S sequencing	A new statement has been included in the Discussion referencing the CLSI guidelines (reference 24). Our recent comprehensive CMR review has also been cited (new reference 33). See page 16, Lines 331-335	A full description of the performance of 16S by clinical microbiology laboratories including the limitations and pitfalls is contained in our CMR review.
	Minor comments – address typos and grammatical errors as listed	All the concerns have been addressed through corrections to the manuscript	
	Please stick to either proteomics or MALDI-TOF throughout the text	All uses of proteomics have been changed to MALDI-TOF throughout	
	Revise Line 68	See the revised statement on page 4, Lines 70-73	
	Laboratory Setting: removed this description	A sentence was added to the Discussion; see page 17, 340-342	Laboratory Setting description has been retained in Methods as we feel it is important to provide this for readers to understand the study context
	Revise Line 114-116	See revision on page 6, lines 120-121	
	Italicize all genus/species names in the text and Tables/Figures; check spacing, update nomenclature and include both the old and new names	Organism names have been checked and revised as recommended throughout the text and illustrations.	

	where required.		
	Line 220 – a (is missing	See page 12, Line 242 – this has been corrected	

Re: Spectrum02545-24R2 (Multi-Year Comparison of VITEK® MS performance for identification of rarely encountered pathogenic gram-positive organisms (GPOs) in a large integrated Canadian healthcare region.)

Dear Dr. Deirdre L. Church:

Your manuscript has been accepted, and I am forwarding it to the ASM production staff for publication. Your paper will first be checked to make sure all elements meet the technical requirements. ASM staff will contact you if anything needs to be revised before copyediting and production can begin. Otherwise, you will be notified when your proofs are ready to be viewed.

Sincerely,
Rebecca Yee
Editor
Microbiology Spectrum